# Fine-Tuning of Alanyl-tRNA Synthetase Quality Control Alleviates Global Dysregulation of the Proteome

**DOI:** 10.3390/genes11101222

**Published:** 2020-10-18

**Authors:** Paul Kelly, Arundhati Kavoor, Michael Ibba

**Affiliations:** 1The Ohio State University Molecular, Cellular and Developmental Biology Program, The Ohio State University, Columbus, OH 43210, USA; kelly.871@buckeyemail.osu.edu (P.K.); kavoor.1@buckeyemail.osu.edu (A.K.); 2Center for RNA Biology, The Ohio State University, Columbus, OH 43210, USA; 3Department of Microbiology, The Ohio State University, 318 West 12th Avenue, Columbus, OH 43210, USA; 4Schmid College of Science and Technology, Chapman University, Orange, CA 92866, USA

**Keywords:** aminoacyl-tRNA synthetases, tRNA, translational fidelity

## Abstract

One integral step in the transition from a nucleic acid encoded-genome to functional proteins is the aminoacylation of tRNA molecules. To perform this activity, aminoacyl-tRNA synthetases (aaRSs) activate free amino acids in the cell forming an aminoacyl-adenylate before transferring the amino acid on to its cognate tRNA. These newly formed aminoacyl-tRNA (aa-tRNA) can then be used by the ribosome during mRNA decoding. In *Escherichia coli*, there are twenty aaRSs encoded in the genome, each of which corresponds to one of the twenty proteinogenic amino acids used in translation. Given the shared chemicophysical properties of many amino acids, aaRSs have evolved mechanisms to prevent erroneous aa-tRNA formation with non-cognate amino acid substrates. Of particular interest is the post-transfer proofreading activity of alanyl-tRNA synthetase (AlaRS) which prevents the accumulation of Ser-tRNA^Ala^ and Gly-tRNA^Ala^ in the cell. We have previously shown that defects in AlaRS proofreading of Ser-tRNA^Ala^ lead to global dysregulation of the *E. coli* proteome, subsequently causing defects in growth, motility, and antibiotic sensitivity. Here we report second-site AlaRS suppressor mutations that alleviate the aforementioned phenotypes, revealing previously uncharacterized residues within the AlaRS proofreading domain that function in quality control.

## 1. Introduction

Across all domains of life, protein-coding sequences have evolved to maintain activity of essential biological processes. Specific residues or modular domains with high evolutionary conservation within a polypeptide sequence can provide insight into regions of functional essentiality of a given protein [1,2]. Because of this, mechanisms have also evolved to minimize errors in protein synthesis [3].

Translational fidelity is maintained at several distinct stages of protein synthesis, with one of the key steps being the accuracy of aminoacyl-tRNA synthesis by aaRSs. AaRSs are responsible for pairing free amino acids in the cell to their cognate tRNAs, with the product then released for participation in ternary complex formation with elongation factor and GTP [4]. This complex is then recruited to the A-site of the ribosome to facilitate peptide bond formation. Most organisms encode at least one distinct aaRS gene for each of the twenty proteinogenic amino acids in the cell. In complex with ATP, aaRSs bind to free cognate amino acids leading to the synthesis of an aminoacyl-adenylate. This activated amino acid will then be transferred to the 3′ end of the tRNA to be released in the cell for translation. AaRSs have evolved mechanisms to prevent the accumulation of mis-aminoacylated tRNAs. Half of the aaRSs in *Escherichia coli* are able to occlude non-cognate amino acid activation in the primary active site; however, the other ten aaRS will mis-activate non-cognate substrates [5]. To prevent non-cognate aa-tRNAs from being released into the cell, many aaRSs have evolved secondary mechanisms to hydrolyze non-cognate products. Several synthetases, including isoleucyl-tRNA synthetase have evolved hydrolytic activity against non-cognate amino acids at the amino acid activation site. Upon non-cognate adenylate recognition, the amino acid and AMP will be hydrolyzed and released back into the cell [6]. In contrast, several aaRS including, phenylalanyl-tRNA synthetase (PheRS) [7], threonyl-tRNA synthetase (ThrRS) [8], and alanyl-tRNA synthetase (AlaRS) [9] encode a distinct secondary active site which contains proofreading activity against non-cognate amino acid-containing substrates after the amino acid is transferred to the 3′ end of the tRNA.

While considerable work has been devoted to characterizing the enzymatic activity of aaRSs, the exploration of the role of these enzymes in vivo has only been recently investigated [10,11,12,13]. It was previously observed that an alanine substitution of an essential cysteine in the AlaRS proofreading domain leads to mis-aminoacylation of serine on tRNA^Ala^ in vitro [9]. *E. coli* with the same substitution (AlaRS C666A) show defects in maintaining proteome homeostasis in *E. coli* [13]. To expand on these efforts, we sought to identify suppressors of the AlaRS C666A phenotype in hopes of further characterizing the mechanisms by which AlaRS fidelity promotes cellular homeostasis. In this work, we identify and characterize two additional residues in *E. coli* AlaRS that play roles in enzymatic activity. All of the novel suppressors appear to modulate AlaRS fidelity through unique and distinct mechanisms. Furthermore, the observations presented herein suggest that a slight increase in AlaRS proofreading activity may be sufficient to alleviate the AlaRS C666A-associated physiological defects. More broadly, this work demonstrates the utility of applying genetic approaches to further elucidate novel mechanistic features of aaRSs.

## 2. Materials and Methods

### 2.1. General Methods

Lysogeny broth (LB) was used for all growth experiments in rich media. M9 media was prepared for the minimal media experiment supplemented with exogenous serine [13]. SOB and SOC media were used for CRISPR/Cas9-mediated strain engineering. When applicable, antibiotics were supplemented at the following concentrations: kanamycin, 25 μg/mL; ampicillin, 100 μg/mL for selection and 20 μg/mL for mistranslation reporter; chloramphenicol 30 μg/mL; spectinomycin 50 μg/mL. All DNA oligonucleotides were synthesized by Sigma Aldrich (St. Louis, MO, USA).

### 2.2. Suppressor Mutant Characterization

Suppressor mutant colonies were isolated and genomic DNA was collected using the Wizard Genomic DNA Purification Kit (Promega, Madison, WI, USA) following manufacturer recommendations. For whole-genome sequencing analysis, 50 ng of genomic DNA was sheared using a Covaris E220 ultrasonicator (Woburn, MA, USA), and sequencing libraries were subsequently prepared using the KAPA Hyper Prep Kit (KAPA Biosystems, Basel, Switzerland). Sequencing was performed on an Illumina HiSeq4000 sequencer (Illumina Inc., San Diego, CA, USA) following the PE150 protocol. Sequencing data were uploaded to the public server at usegalaxy.org and sequencing fragments ranging between 35–151 bp were analyzed using the Galaxy web platform [14]. AlaRS C666A suppressor mutations were also identified using Sanger sequencing of PCR-amplified fragments of the AlaRS editing domain.

AlaRS protein sequences were aligned using the EMBL-EBI Clustal Omega multiple sequence alignment tool [15] and visualized using Jalview [16]. Positions of the suppressor-substituted amino acids were modeled using Phyre2 [17] and visualized using PyMOL software.

### 2.3. Strain Construction

The AlaRS C666A suppressor in *alaS* at nucleotide position G1651T (encoding AlaRS D551Y) was made in an isogenic MG1655 strain using the “gene gorging” method as has previously been described [13]. Suppressor mutations encoding substitutions at R561 were also generated in isogenic MG1655 derivative strains. These four strains (single R561C and R561S variants and double R561C/S C666A variants) were constructed using CRISPR/Cas9 genome engineering [18,19]. Following the cloning strategies described by Reisch and colleagues, the suppressor strains were constructed and validated by Sanger sequencing of the AlaRS proofreading domain amplicon.

### 2.4. Growth Analysis

For all growth experiments, saturated overnight cultures were normalized and back-diluted to an OD_600_ of 0.05 in the respective experimental media. Cultures were then incubated at either 37 °C or 42 °C and OD600 values were monitored using a CO8000 cell density meter (WPA) at 30′ time intervals. The data plotted are the averages of three biological replicates with error bars indicating the standard deviation of the replicates.

### 2.5. Motility Assays

To monitor changes in *E. coli* motility, swimming plates were prepared using LB media supplemented with 0.2% agar. Saturated overnight cultures were normalized to an OD_600_ of 0.5 and 0.5 µL of cells were spotted on the agar plates. The swimming plates were incubated at 37 °C for 8 h prior to imaging. *E. coli* motility was quantified by measuring the swimming diameter using ImageJ.

### 2.6. Antibiotic Sensitivity

Sensitivity to antibiotic exposure was monitored by streaking bacterial lawn cultures on LB agar plates and co-incubation with Oxoid antibiotic disks (Thermo Fisher, Waltham, MA, USA). Plates were incubated overnight at 37 °C and the sensitivities to ertapenem and polymixin B were quantified by measuring the diameter of antibiotic clearing using ImageJ.

### 2.7. In vivo Mistranslation Reporter

The construction and utility of the pLK-Amp S68A mistranslation reporter have been previously described [13]. In brief, β-lactamase, a gene product used for conferring ampicillin resistance, contains an essential serine residue at position S68. By mutating the gene to encode for an alanine residue at the 68 position, active β-lactamase translation is dependent upon mistranslation of the alanine to serine at that position. The aforementioned mistranslation event can be monitored by the ability of microbial growth in the presence of ampicillin. Reporter plasmids containing either wild-type or mutant β-lactamase genes were transformed into all of the MG1655 AlaRS C666A suppressor strains. The resulting strains were struck onto LB agar plates containing ampicillin and growth was monitored after 48 h of growth at 37 °C.

### 2.8. Preparation of Recombinant Protein and In Vitro Transcribed tRNA^Ala^

Genes encoding wild-type AlaRS or the AlaRS C666A variant were cloned into pET21b at NdeI and XhoI restriction sites. The aforementioned cloning strategy generated an in-frame C-terminal His-tag for metal affinity purification as previously described [13]. To characterize the biochemical activity of the suppressor mutations, site-directed mutagenesis was used to create single and double mutant AlaRS expression constructs. Active enzyme concentrations were determined by active site titration [13,20].

In vitro transcribed *E. coli* tRNA^Ala^ was prepared by slow cooling partially overlapping synthetic DNA oligonucleotides which encode for a T7 promoter and the most abundant tRNA^Ala^ isoacceptor in *E. coli* (genes: *alaT*, *alaU*, and *alaV*). The annealed fragments were then ligated and cloned into EcoRI and XbaI restriction sites in pUC18. Plasmid containing the *E. coli* tRNA^Ala^ gene was used as a template for PCR amplification of the T7 promoter and tRNA. The PCR-amplified product was subsequently used for T7 runoff transcription. In vitro transcription was performed as previously described [21] and the tRNA was purified using anion exchange chromatography.

### 2.9. Proofreading Activity

To determine if the AlaRS suppressor variants led to changes in mis-aminoacylation in vitro, all single and double mutant AlaRS variants were used for in vitro mis-serylation experiments. To monitor mis-aminoacylation, 5 μM AlaRS was incubated with 5 μM tRNA^Ala^, 750 μM [^3^H]-Ser, aminoacylation buffer (100 µM HEPES pH 7.2, 30 mM KCl, and 10 mM MgCl_2_), and initiated with 8 mM ATP. Mis-serylation reactions were carried out for at 37 °C 15 min and radiolabeled signal was determined by TCA precipitation and subsequent quantification by scintillation counting.

Beyond mis-aminoacylation, *trans* proofreading activity was monitored to determine if the suppressor mutants altered aminoacyl-tRNA re-binding and subsequent proofreading. To generate pre-formed aa-tRNA^Ala^, recombinant EF-Tu was first activated (EF-Tu expression construct was generously provided by Dr. Kurt Fredrick, Ohio State University). EF-Tu activation was performed by incubating 50 μM EF-Tu with 50 mM Tris pH 7.8, 100 μM DTT, 68 mM KCl, 6.7 mM MgCl_2_, 2.5 mM phosphoenol pyruvate, 30 μg pyruvate kinase, and 500 μM GTP. EF-Tu activation reactions were incubated at 37 °C for 30 min and used immediately for mis-aminoacylation. Pre-formed Ser-tRNA^Ala^ was subsequently generated by incubating 10 μM tRNA^Ala^ with 950 μM (^3^H)-Ser, 5 μM activated EF-Tu, 5 μM *E. coli* AlaRS C666A, 8 mM ATP, and aminoacylation buffer. Cognate Ala-tRNA^Ala^ was pre-formed in buffering conditions as described above and with 150 μM (^14^C)-Ala, 200 nM *E. coli* AlaRS, and 5 μM tRNA^Ala^. Aminoacylation reactions were incubated for 1 h at 37 °C and subsequently quenched by adding an equal volume of acid phenol chloroform. Following acid phenol chloroform extraction, aa-tRNA^Ala^ was EtOH precipitated, before finally being re-suspended in 100 mM sodium acetate pH 4.5. To monitor aa-tRNA^Ala^ deacylation activity, 150 nM AlaRS variants were incubated with aminoacylation buffer and pre-formed aa-tRNA^Ala^. Aliquots of the reaction were quenched on 5% TCA presoaked filter paper at various time points between 0 and 20 min. Quenched reactions were then washed 3x in 5% TCA, EtOH, dried, and quantified using scintillation counting. Percent deacylation was determined by the decrease in radiolabeled signal relative to T_0_.

To monitor ATP consumption, reactions including: 2 μM tRNA^Ala^, 1× aminoacylation buffer, 1 μM AlaRS variants, 0.2 U PPiase, 3 mM (^32^P – γ) ATP, and either 10 mM alanine or 100 mM serine were incubated at 37 °C. At time points ranging between 3′–30′, aliquots of the reaction mix were removed and quenched in an equal volume of glacial acetic acid. To monitor the consumption of ATP, migration of the quenched reactions were monitored by thin-layer chromatography on PEI cellulose plates. Plates were developed in dipotassium phosphate and imaged using phosphor imaging.

### 2.10. Pyrophosphate Exchange

Pyrophosphate (PPi) exchange assays were performed at 37 °C in HKM buffer (100 mM Na-HEPES, pH 7.2, 30 mM KCl, 10 mM MgCl_2_, 2 mM NaF) containing 2 mM ATP, 2 mM (^32^P) PPi (2–4 cpm/pmol) and 10 nM of the purified enzyme. The concentrations of the substrates were varied from 0–1 mM for alanine and 0–50 mM for serine. The reactions were quenched at the selected timepoints by aliquoting 25 µL of the reaction into 970 μL of a charcoal solution (1% charcoal, 5.6% HClO_4_, and 75 mM PPi). The quenched solution was then added onto 3 MM Whatman filter discs and vacuum filtered with three 5 mL washes of water. The discs were then dried and the radioactivity was determined by liquid scintillation counting. Results were fit to a Michaelis-Menten curve using GraphPad Prism to determine the steady-state activation kinetics.

## 3. Results

### 3.1. Identification of Suppressor Mutations That Alleviate the AlaRS C666A Growth Defect

It has been previously observed that perturbation of AlaRS proofreading in *E. coli* leads to growth defects when compared to wild-type *E. coli* [13]. To determine if secondary mutations in the *E. coli* genome could suppress the growth defects associated with a deficiency in AlaRS proofreading, suppressor mutants that restore near-normal growth compared to their proofreading-deficient strain were screened for. Due to the severe growth defects associated with AlaRS-mediated mistranslation, suppressor mutations were anticipated to act in reducing the mistranslational load on the proteome. It was speculated that this suppression could possibly occur by increasing the thermodynamic barrier for EF-Tu discrimination and thus preventing mis-acylated tRNA accumulation into an active ternary complex [22,23,24]. Additionally, the *E. coli* protein ProXP-ST1 was previously shown to have activity against Ser-tRNA^Ala^ in vitro [25]. Therefore it was possible that mutations in the *yeaK* gene (gene encoding ProXP-ST1) could increase ProXP-ST1 proofreading activity on mis-serylated tRNA^Ala^ substrates.

Suppressor colonies of the MG1655 AlaRS C666A strain accumulated on LB agar plates without the need for additional mutagenic manipulation. One initial wild-type-like colony was selected from the mixed population, colony purified, and genomic DNA (gDNA) was extracted. The isolated gDNA was further subjected to whole-genome sequencing (WGS) for variant analysis. When compared to another control sample that was also analyzed by WGS, only one additional mutation was identified in the suppressor mutant. The identified single nucleotide polymorphism (SNP) was a G1651T mutation in the *alaS* gene that when translated, caused a D551Y substitution in AlaRS. Having identified second-site *alaS* suppressors, two additional suppressor mutant colonies were isolated from LB agar plates. Rather than subjecting the two additional mutants to WGS analysis, gDNA was extracted from these mutants and the DNA was used as the template for PCR to amplify the *alaS* gene. This *alaS* amplicon was used as the template for a Sanger sequencing reaction using a primer with homology to *alaS*. Sanger sequencing of the two additional suppressors reported that both of the mutants also contained secondary mutations in the *alaS* gene, with both SNPs occurring at nucleotide position C1681, ultimately resulting in two different substitutions at R561, R561C, and R561S.

While the *alaS* gene encodes one complete monomeric unit of AlaRS, this gene contains at least three characterized domains responsible for distinct enzymatic activity including, amino acid activation and tRNA aminoacylation [26,27], proofreading [9], and a C-terminal C-Ala domain required for oligomerization that also participates in aminoacylation [28,29]. The three suppressor mutations all occur in the proofreading domain of AlaRS (Figure 1A). To date, these two amino acid positions have yet to be characterized or predicted to participate in any proofreading activity. The coordination of the proofreading domain active site has previously been explored and has indicated that residues involved in mis-acylated tRNA hydrolysis are evolutionarily conserved (e.g., C666 in *E. coli*) [30]. To determine if the two amino acid residues identified in the suppressor screen are also conserved, AlaRS proofreading domains were aligned across several prokaryotic and eukaryotic model systems. With the exception of the closely related *Salmonella enterica*, these amino acid positions do not appear to be evolutionarily conserved (Figure 1B). Currently, no structural data for the *E. coli* AlaRS proofreading domain exists. Using Phyre2 modeling predictions, the position of the suppressor residues were mapped on the AlaRS structure. Based on these predictions, both residues, D551 and R561 are positioned in potentially productive regions in the AlaRS proofreading domain for tRNA body interactions, but likely unable to directly participate in mis-aminoacyl-tRNA hydrolysis (Figure 1C). These preliminary observations provided the first evidence of second-site AlaRS suppressors that alleviated the growth defect associated with AlaRS mistranslation in *E. coli*.

### 3.2. Second-Site AlaRS Mutations Alleviate the Slow-Growth AlaRS C666A Phenotype

While the sequencing data indicated second-site mutations in the *alaS* gene were likely responsible for the suppression of the AlaRS C666A slow-growth phenotype, it was possible that other mutations in the suppressor strains may have been missed by our sequencing analysis and the second-site mutations were a coincidence but not responsible for the suppression. To confirm that the three identified mutants were acting as the MG1655 AlaRS C666A suppressors, the three mutations were made in isogenic MG1655 and MG1655 AlaRS C666A strains, creating single and double mutants for the suppressors. One possibility for these mutants was the potential for the second-site mutants acting as compensatory mutations for the AlaRS C666A substitution, while single mutants alone may elicit a separate growth defect.

Upon generation of the six isogenic strains, growth of all of the strains was monitored in LB liquid media at 37 °C. This growth condition has previously been used to highlight the AlaRS C666A growth defect [13]. Interestingly, all single and double mutant suppressor strains grew as well as wild-type *E. coli* (Figure 2A). This observation indicated that suppressor mutations identified from the various sequencing analyses were acting as suppressors. Furthermore, substitutions at D551 or R561 alone do not cause any change in growth at 37 °C in rich media. As previously eluded to, perturbation to AlaRS proofreading leads to defects in growth in rich media. It was further shown that this defect is serine dependent, presumably through elevated levels of Ala to Ser mistranslation [9,13]. To determine if the suppressor mutants also exhibit a serine-dependent growth defect that may have been masked in rich media, the suppressor mutants were grown in minimal media supplemented with exogenous serine, and their growth was monitored. Despite the excess non-cognate amino acid stress, the single and double suppressor mutants were able to grow just as well as wild-type *E. coli* (Figure 2B), which may suggest that the level of serine mistranslation was reduced in the presence of the second-site substitutions.

### 3.3. AlaRS C666A Suppressors Do Not Prevent Serine Mistranslation

To directly observe the potential for Ala to Ser mistranslation in the suppressor strains, a recently described mistranslation reporter [13] was applied to all of the single and double mutants. The mistranslation reporter utilizes a mutated β-lactamase gene in which an essential serine residue has been mutated to encode an alanine [31]. Under conditions of high AlaRS fidelity, the mutant β-lactamase product will be translated and the cells will be unable to grow on β-lactam antibiotics (e.g., ampicillin). If Ala to Ser mistranslation is occurring, the wildtype β-lactamase protein will be made and the cells will grow in the presence of the antibiotic. This reporter has recently been used to show that in the AlaRS C666A variant strain, Ala to Ser mistranslation is occurring in vivo [13]. Upon transformation of the reporter and plating of the suppressor strains, all variants that contained the C666A allele were able to grow on ampicillin (Figure 3). This observation suggests that the C666A substitution is sufficient to cause protein mistranslation even in the presence of additional second-site suppressors. None of the single AlaRS suppressor variants were able to grow in the presence of antibiotics suggesting that these substitutions alone do not negatively affect the hydrolytic proofreading activity of these enzymes in vivo. One limitation of this qualitative assay is the inability to discern quantifiable changes in levels of mistranslated proteins. It is possible that despite the accumulation of mistranslated proteins in the second-site suppressors, the levels of mistranslation are still reduced compared to the single AlaRS C666A variant.

### 3.4. Phenotypes Associated with AlaRS-Mediated Proteome Dysregulation Are Alleviated in with the Second-Site Suppressor Mutations

Previous efforts studying the effect of AlaRS mistranslation in *E. coli* identified that not only was protein mistranslation occurring, but the accumulation of mistranslated proteins led to dysregulation of the *E. coli* proteome [13]. This observation suggested that regardless of the individual protein that may contain a misincorporated amino acid, the Ala to Ser substitution itself was not tolerated globally and caused activation of several stress response cascades that could be reproducibly observed with population-level analyses. The first indication that protein regulation may be altered in the *E. coli* AlaRS strain was the observation that in the absence of AlaRS proofreading, *E. coli* exhibited an accentuated slow growth phenotype when grown in rich media at 42 °C. The delayed growth response to the heat stress potentially suggested that these AlaRS proofreading-deficient cells were unable to mount an adequate response to thermal stress. Total proteome analysis of wild-type and AlaRS C666A strains further revealed that in the absence of AlaRS proofreading, more than 800 proteins were differentially enriched or under-represented [13].

Despite the observation that serine mistranslation was still occurring in the suppressor mutants, it was of interest to determine if the phenotypes associated with the proteome dysregulation in the AlaRS C666A strain were still present in the suppressor strains. To determine if the suppressor strains were able to respond to a shift in thermal stress, the suppressor strains were grown in LB media at 42 °C. As had been previously observed [13], the AlaRS C666A strain has an elevated growth sensitivity at 42 °C compared to 37 °C (Figure 2A and Figure 4A). However, none of the single or double suppressor mutants elicited any deviation in growth when compared to wild-type *E. coli* (Figure 4A). This observation suggested that the dysregulation associated with the single AlaRS C666A allele was not present in the suppressor strains.

From the original total proteome analysis, several distinct physiological pathways were identified to be enriched or under-represented. In addition to many other biological processes, there appeared to be a defect in motility and membrane integrity that was further assayed for and confirmed using direct phenotypic analyses [13]. To further support that the global dysregulation was alleviated in the AlaRS C666A suppressors, all strains were plated on LB agar swimming media, and motility was monitored. The only strain to exhibit a statistically significant change in motility was the single AlaRS C666A mutant, as had previously been observed. All of the suppressors were able to swim as well as wild type (Figure 4B). Perturbations to membrane integrity were monitored by antibiotic disk diffusion assays. Once again, all suppressor strains were equally sensitive to both ertapenem and polymixin B as wild-type *E. coli*, whereas the AlaRS C666A variant was significantly more sensitive to antibiotic exposure (Figure 4C). The results presented herein highlight that Ala to Ser mistranslation is not sufficient to cause global proteome dysregulation. Having demonstrated that the second-site AlaRS suppressors were able to not only alleviate the slow growth defect associated with the AlaRS C666A variants (Figure 2) but also likely restore the global proteome homeostasis (Figure 4) but still not prevent Ala to Ser mistranslation (Figure 3), the mechanism for this suppression still remained unclear.

### 3.5. In Vitro Characterization of the AlaRS C666A Suppressors

Having previously demonstrated that the AlaRS C666A suppressors do not prevent mistranslation in vivo, biochemical analyses were performed to potentially identify differences in the formation of the Ser-tRNA^Ala^ substrates in vitro. Recombinant AlaRS variants were expressed and purified from *E. coli* and in vitro transcribed *E. coli* tRNA^Ala^ was used as a tRNA substrate for all cognate and non-cognate experimentation. To determine if there were any differences in non-cognate Ser-tRNA^Ala^ formation, in vitro mis-aminoacylation experiments were performed using all of the AlaRS variants. The AlaRS C666A editing-defective variant was able to mis-serylate tRNA^Ala^ as expected, and all of the AlaRS suppressor enzymes were also able to mis-aminoacylate tRNA^Ala^ (Figure 5A). While the AlaRS suppressor variants were able to mis-serylate tRNA^Ala^, AlaRS C666A R561C did exhibit a ~50% reduction in Ser-tRNA^Ala^ product formation (Figure 5B). Consistent with observations using the in vivo mistranslation reporter, this in vitro analysis suggests that the AlaRS suppressor variants are unable to prevent mis-aminoacylation but a substitution of R561C can potentially lead to an overall decrease in mis-serylated tRNA formation in the cell.

In addition to the direct contribution of Ser-tRNA^Ala^ product formation, it was possible that the mechanism for the suppressors could be occurring due to changes in Ser-tRNA^Ala^ proofreading. Pre-formed Ser-tRNA^Ala^ was incubated with all of the AlaRS variants and deacylation was monitored over time. None of the second-site suppressors were able to proofread the mis-serylated tRNA product similar to wild-type AlaRS levels (Figure 5C). The AlaRS R561S variant enzyme exhibited elevated levels of cognate Ala-tRNA^Ala^ deacylation, with the percent of deacylated substrates reaching ~40% after 30 min (Figure 5D). Upon further inspection of the initial rates of Ser-tRNA^Ala^ deacylation, two of the single mutant suppressors, AlaRS D551Y and AlaRS R561S appear to be proofreading faster than the wild-type enzyme (Figure 5E). Furthermore, evaluation of the initial deacylation rates for the double mutants suggests that AlaRS D551Y C666A and the AlaRS R561S C666A enzymes have slightly elevated proofreading activities compared to the editing-defective AlaRS C666A enzyme (Figure 5F). An additional observation made from the in vitro deacylation experiment is that the single AlaRS R561C variant was unable to deacylate Ser-tRNA^Ala^ in trans. As previously described, the AlaRS R561C variant was unable to form mis-serylated tRNA^Ala^ product in vitro, suggesting that an additional conformational change may have occurred in this enzyme leading to greater fidelity during initial serine activation that no longer requires proofreading activity in the editing domain. This hypothesis is further supported by results from the in vivo reporter experiment, which did not reveal Ala to Ser mistranslation in the MG1655 AlaRS R561C mutant background (Figure 3).

A secondary measure for determining aaRS proofreading activity is through the monitoring of ATP futile cycling. As has been previously described, the first step in aa-tRNA formation is the activation of free amino acid with ATP to form an aminoacyl-adenylate. During standard cognate aa-tRNA reactions, one molecule of ATP is activated per one aa-tRNA formed, ultimately reaching equilibrium when all tRNA substrates are aminoacylated. Alternatively, when aaRS proofreading is occurring, the mis-activated substrate will be hydrolyzed and can subsequently be re-activated, leading to iterative rounds of ATP consumption.

Observations from monitoring ATP cycling suggest that AlaRS R561S was the most proofreading-proficient AlaRS variant as ATP consumption levels surpassed those of the wild-type enzyme (Figure 5G). Furthermore, AlaRS R561S displayed increased rates of ATP consumption when incubated with cognate alanine, consistent with observations that R561S leads to cognate Ala-tRNA^Ala^ deacylation (Figure 5H). While the AlaRS D551Y enzyme displayed the highest rate of Ser-tRNA^Ala^ deacylation in trans, similar levels of ATP consumption were observed when compared to wild-type, suggesting an imbalance in the rate of deacylation and re-activation. Finally, enzymes containing the R561C variant had the lowest rate of ATP consumption, which is consistent with deacylation studies suggesting that these enzymes are unable to proofread Ser-tRNA^Ala^.

To further characterize the AlaRS editing suppressors, amino acid activation was monitored using a pyrophosphate (PPi) exchange. This assay enables the determination of amino acid activation kinetics by measuring the accumulation of (^32^P)-ATP in a reaction containing excess (^32^P)-PPi. PPi exchange not only facilitates the determination of aminoacyl-adenylate kinetics, it can also be used to highlight changes in enzyme specificities for different substrates, such as cognate alanine or non-cognate serine.

Observations from the pyrophosphate exchange assay indicate that the single AlaRS R561C variant shows increased specificity against serine activation (Table 1). This is consistent with previous observations where the AlaRS R561C variant was unable to hydrolyze Ser-tRNA^Ala^ in trans (Figure 5C) while also being unable to form Ser-tRNA^Ala^ (Figure 5B). The AlaRS suppressor C666A R561C also shows an increased specificity against serine activation as compared to AlaRS C666A (Table 1). This increased serine discrimination by the R561C suppressor may explain the ~50% reduction in Ser-tRNA^Ala^ formation (Figure 5B). In contrast, the AlaRS C666A R561S mutant shows similar specificity against serine activation as compared to AlaRS C666A (Table 1). This indicates that the R561S mutation has no effect on amino activation but instead increases overall proofreading as evidenced by the deacylation studies (Figure 5C,D) and ATP futile cycling (Figure 5G,H). Finally, the single AlaRS D551Y variant and the suppressor AlaRS C666A D551Y show reduced specificity against serine activation as seen in Table 1. However, the AlaRS D551Y variant showed the highest rate of deacylation for Ser-tRNA^Ala^ (Figure 5E). Thus, the D551Y substitution is likely productive for Ser-tRNA^Ala^ proofreading but not for preventing serine misactivation.

### 3.6. AlaRS R561C Reduces Ser-tRNA^Ala^ Formation

Biochemical characterization of the AlaRS R561C variants suggests this enzyme has an overall reduction in the potential to form Ser-tRNA^Ala^, as is evident in Figure 5B. This reduced potential to form Ser-tRNA^Ala^ is likely due to an increased specificity against serine during amino acid activation, as seen in Table 1. Interestingly, the R561C substitution led to a reduction in overall proofreading activity even with C666 present (Figure 5C). This interpretation is consistent with observations monitoring ATP consumption (Figure 5G). Taken together, the biochemical analyses suggest that AlaRS R561C is leading to increased discrimination of serine, independent of AlaRS proofreading. While it is yet to be explored, the R561C substitution may cause disruption to wild-type AlaRS disulfide bond formation, leading to changes in enzyme activation and proofreading.

### 3.7. AlaRS R561S Has Elevated Proofreading Activity

In contrast to observations studying the effect of the AlaRS R561C on aaRS discrimination, the AlaRS R561S variant appears to be hyper-active in AlaRS proofreading, as indicated by the increased hydrolytic activity. The AlaRS R561S C666A variant is able to mis-serylate tRNA^Ala^ which is consistent with the observation of Ala to Ser mistranslation in vivo (Figure 3). Furthermore, this enzyme shows no changes in the discrimination of serine during amino acid activation, as compared to AlaRS C666A (Table 1). However, the levels of Ser-tRNA^Ala^ do appear to be slightly reduced compared to the AlaRS C666A single mutant (Figure 5A). The elevated proofreading activity is more obvious when monitoring the deacylation activity directly in editing assays. The R561S variant has elevated rates of Ser-tRNA^Ala^ deacylation compared to wild-type AlaRS (Figure 5E), and the R561S C666A enzyme has elevated proofreading activity compared to editing-defective AlaRS C666A variant (Figure 5F). Consistent with these observations is the elevated levels of ATP consumption in enzymes containing the R561S substitution. The single AlaRS R561S variant had the highest rate of ATP consumption and the R561S C666A variant had elevated ATP consumption compared to the C666A enzyme (Figure 5G). Interestingly, the R561S enzyme had elevated levels of deacylation (Figure 5D) and ATP consumption for cognate Ala-tRNA^Ala^ substrates (Figure 5H). This observation provides the first evidence of a hyper-active proofreading enzyme. Furthermore, the contribution of AlaRS R561S together with the editing-defective C666A substitution may lead to an overall reduction of Ser-tRNA^Ala^ in the cell.

### 3.8. AlaRS D551Y Increases Ser-tRNA^Ala^ Deacylation in Trans

Compared to observations from the R561 variants, the AlaRS D551Y suppressor is unique in that the double mutant suppressor enzyme had no change in Ser-tRNA^Ala^ formation (Figure 5A). The mis-serylation experiments including the AlaRS D551Y C666A enzyme were the most variable of all other enzymes tested. To circumvent this problem, this experimental condition was tested nine total times compared to the three replicates of the other enzymes. After much consideration, we believe that this variability is not an artifact of technical error but rather, suggestive of perturbation to the overall rate of AlaRS activity. Pyrophosphate exchange assays indicate that the AlaRS D551Y C666A enzyme has the lowest specificity against serine during amino acid activation (Table 1). When studying proofreading activity directly, AlaRS D551Y had the fastest rate of Ser-tRNA^Ala^ deacylation of all enzymes tested (Figure 5E). Interestingly, the elevated rate of deacylation was not observed when monitoring the rate of ATP consumption (Figure 5G). Together these results suggest that AlaRS D551Y is able to efficiently deacylate Ser-tRNA^Ala^ in trans. As the D551Y containing-containing enzymes had reduced specificity against non-cognate serine, this suggests that activation is likely not limited, but rather likely due to a perturbation during tRNA transfer. While this discrepancy still remains unclear, the data presented herein do suggest that the AlaRS D551Y substitution likely reduces the overall Ser-tRNA^Ala^ burden caused by the C666A substitution.

## 4. Discussion

### 4.1. Fitness Costs Associated with Translational Errors and Disruption of the Aminoacyl-tRNA Pool Are Amino Acid-Specific

The aforementioned suppressor screen identified an AlaRS allele that leads to increased accuracy against non-cognate aa-tRNA^Ala^ formation. Recombinant AlaRS R561S enzymes displayed elevated rates of non-cognate Ser-tRNA^Ala^ hydrolysis as well as a long-term increase in cognate Ala-tRNA^Ala^ deacylation. This observation provides the first direct evidence of a hyper-accurate AlaRS enzyme, and to our knowledge, the first experimental validation of an aaRS post-transfer proofreading domain with increased hydrolytic activity. Interestingly, the single AlaRS R561S mutant displays no distinguishable phenotypic perturbation compared to wild-type *E. coli*. This observation suggests that perturbation to Ala-tRNA^Ala^ fidelity may be less tolerated than an overall decrease in tRNA^Ala^ aminoacylation levels.

It has previously been noted that in both bacteria and eukaryotes, perturbation to aaRS fidelity not only affects rates of protein mistranslation but can also alter intracellular sensing of nutritional stress and consequently leading to dysregulation of cellular stress [11,12]. AaRSs must maintain sufficient aa-tRNA levels to keep up with the translational demand for cellular protein synthesis and also monitor amino acid availability for nutrient sensing. Both of these roles make aaRSs an important player to maintain and modulate cellular homeostasis. In *E. coli* the accumulation of uncharged tRNA will activate the stringent response facilitated by the RelA-dependent accumulation ppGpp [32,33]. Stringent response activation leads to down-regulation of rRNAs and ribosomal proteins, ultimately slowing down cellular growth [33,34]. Numerous reports have indicated the detrimental cost of AlaRS-mediated translational errors, but here we suggest that those errors are more costly than perturbations to the pool of aa-tRNA^Ala^ formation. We have previously suggested that specific mistranslational errors are more tolerated in the proteome than others [13]. We would now like to expand on that observation and further suggest that not only are specific mistranslational errors more or less costly to the proteome, but defects associated with aa-tRNA formation are also amino acid-dependent as well.

### 4.2. Perturbation of AaRS Fidelity Leads to Rapid Acquisition of the AlaRS C666A Variant

One striking observation from this work was the accumulation of AlaRS C666A suppressors without the necessity for mutagenic manipulation. It has previously been reported that increased levels of mistranslation, either facilitated by an error-prone aaRS [35] or imbalances in initiator tRNA levels [36] lead to increased SOS-dependent mutation sampling. This model proposes that increased levels of mistranslation cause elevated Lon protein levels, ultimately leading to the RecA-dependent degradation of LexA causing SOS induction. The authors of the aforementioned work first noted decreased antibiotic sensitivity to the DNA gyrase inhibitor ciprofloxacin during increased protein mistranslation.

Previous work from our group has shown that defects in AlaRS proofreading lead to increased sensitivity to ciprofloxacin [13]. Similar analysis comparing a proofreading-deficient ThrRS *E. coli* strain suggested a slight trend towards improved sensitivity to ciprofloxacin compared to wild type, although the observed trend was not statistically significant [13]. Together these observations may suggest that while SOS induction may be occurring in response to increased mistranslation, the cost of a specific error may affect the potential benefit of this early stress response induction.

Total proteomic analysis of the *E. coli* AlaRS C666A strain reported a ~1.7× increase in RecA levels compared to wild type [13]. While predicted correlating trends in both Lon and LexA were not observed, it is possible that the observed changes in RecA correspond to other observations of mistranslation causing SOS-dependent mutant sampling. This mechanism may be responsible for the propensity for the AlaRS C666A suppressor to arise that were identified in this report.

### 4.3. Secondary AaRS Proofreading Activities Permit Regulation of Translational Quality Control

Half of the aaRSs in *E. coli* encode for distinct secondary proofreading activities that either hydrolyze mis-activated aminoacyl-adenylates or mis-aminoacylated tRNAs. Perturbation to these activities has indicated a wide array of neutral to deleterious outcomes for cells in vivo [10,13,37]. The extensive evolutionary conservation of aaRS proofreading would suggest an important role for the discriminatory properties of these enzymes that would otherwise not be possible by active site discrimination alone.

In this work, we have identified an AlaRS variant, AlaRS R561C which sufficiently prevents serine mistranslation in vivo but is unable to hydrolyze mis-aminoacylated in vitro and elicits no futile cycling of ATP, consistent with predictions that this enzyme has lost hydrolytic proofreading activity. This observation raises an interesting idea in that why would AlaRSs maintain an energetically-unfavorable proofreading activity when an AlaRS variant can exist that is sufficiently discriminatory and also does not require the consumption of ATP? Of the phenotypes explored in this report, the *E. coli* AlaRS R561C variants displayed no differences compared to the wild-type counterpart *E. coli*. It is possible that there are other physiologically relevant environments that may elicit deviations from that trend however, there is another possibility why aaRS proofreading may be maintained that we would like to propose.

Work from our group and others has continually shown the incredible cost associated with perturbation to AlaRS proofreading [13,37,38,39]. However, thinking about aaRSs more globally, we know that some errors are better tolerated than others [13], and some aaRSs only need proofreading activity against non-proteinogenic substrates [40,41]. There is a growing body of evidence that aaRS proofreading activity can also be modulated by environmental changes. Both ThrRS [42] and PheRS [21] alter proofreading activity upon oxidative stress, both by decreasing and increasing their discriminatory capacities respectively. Furthermore, ThrRS was recently shown to decrease proofreading activity upon post-translational acetylation [43]. These observations suggest a potential role for maintaining secondary proofreading activities as opposed to encoding for ubiquitously sufficient discriminatory activities in the primary active site. The maintenance of hydrolytic proofreading activities allows the opportunity to modulate enzyme discrimination during environmental or regulatory changes. All together suggesting that while it is possible for an aaRS to maintain active site discrimination without the necessity for distinct proofreading mechanisms, it abolishes the opportunity for modulation of proofreading activities and therefore has potentially not been selected for evolutionarily.

## 5. Conclusions

Aminoacyl-tRNA synthetases are essential enzymes that help maintain the speed and accuracy of the cellular proteome. It has previously been observed that defects in aaRS fidelity can lead to a wide array of cellular defects, likely due to the cost of substitution-specific effects on protein dynamics. Of the aaRSs that have been explored to date, defects in AlaRS proofreading appear to be one of the most severe, with perturbations to AlaRS fidelity causing global dysregulation of the *E. coli* proteome. In this work we used a genetic screen to identify mutations that would suppress the deleterious AlaRS proofreading-deficient phenotypes observed in *E. coli*. Through these efforts we identified three second-site suppressors in the AlaRS proofreading domain that alleviate the growth defects associated with AlaRS errors, but interestingly do not completely prevent protein mistranslation. Subsequent biochemical analysis highlighted that all three of the suppressors work through unique mechanisms to reduce the overall Ser-tRNA^Ala^ burden in the cell and these subtle changes are sufficient to restore proteome regulation in *E. coli*.

## Figures and Tables

**Figure 1 genes-11-01222-f001:**
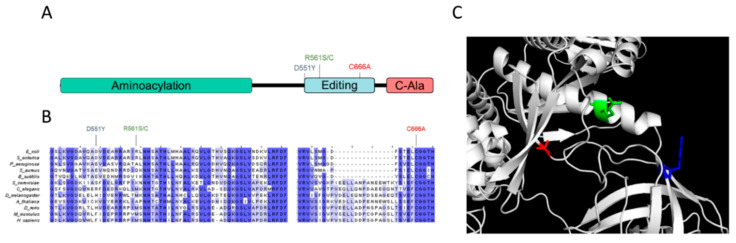
AlaRS C666A suppressors map to the AlaRS proofreading domain. (**A**) Suppressor mutations map to the AlaRS proofreading domain. (**B**) The sites for suppressor mutants do not appear to be conserved across all domains of life. (**C**) Suppressor variants were modeled on the AlaRS proofreading domain using Phyre2 (96% of residues modeled at >90% accuracy).

**Figure 2 genes-11-01222-f002:**
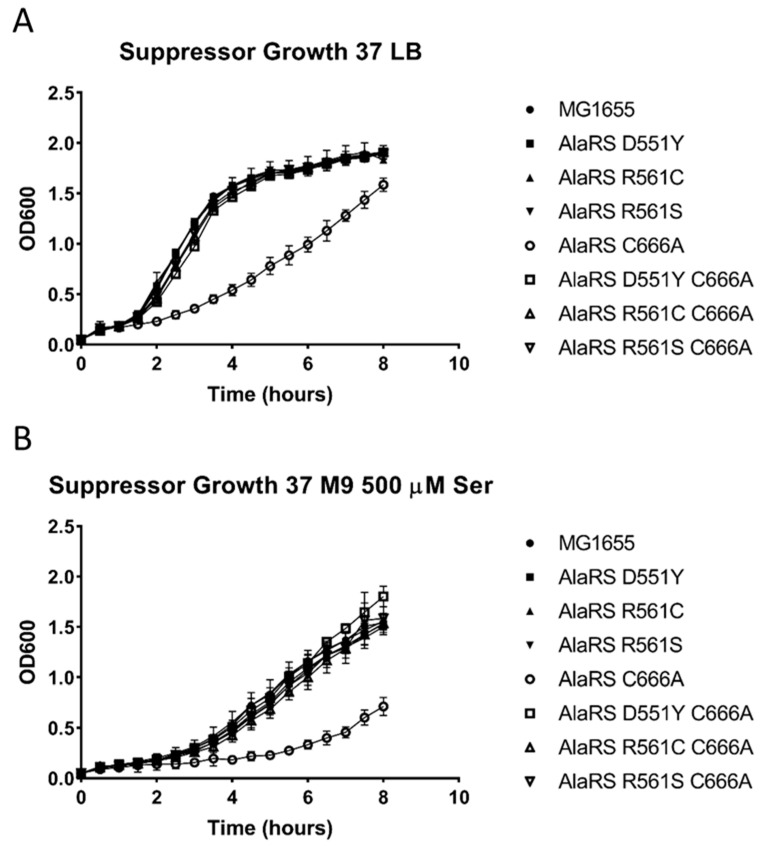
Suppressor mutants alleviate AlaRS C666A-associated growth defects. Suppressor mutants alleviated the AlaRS C666A growth defects in (**A**) LB (Lysogeny broth) and (**B**) M9 minimal media supplemented with non-cognate serine.

**Figure 3 genes-11-01222-f003:**
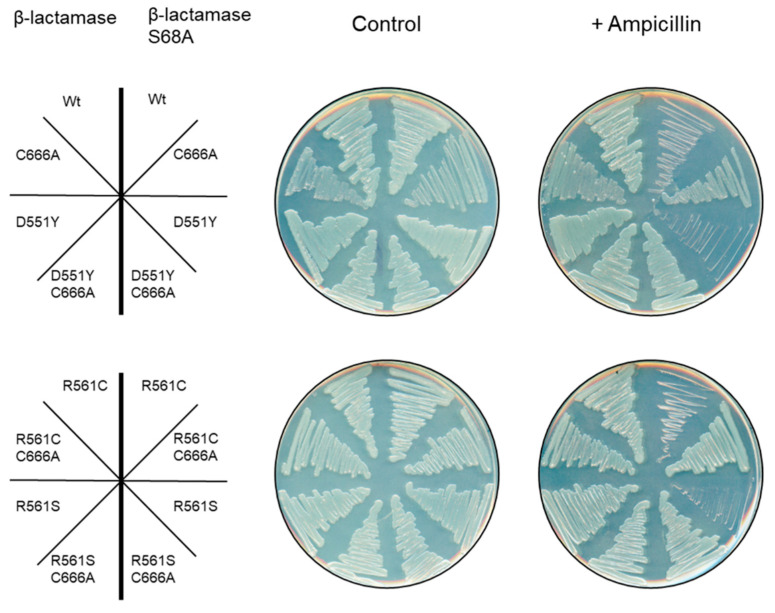
AlaRS C666A second-site suppressors do not prevent mistranslation. Serine mistranslation was observed in the AlaRS C666A suppressors using a β-lactamase S68A mistranslation reporter.

**Figure 4 genes-11-01222-f004:**
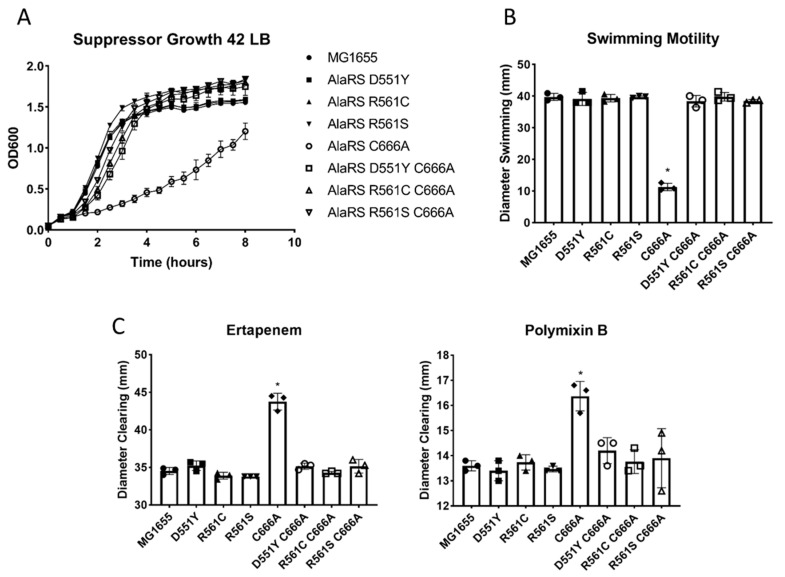
Suppressor mutants alleviate AlaRS C666A-associated phenotypes. Second-site suppressors alleviated the (**A**) heat stress, (**B**) swimming, and (**C**) antibiotic sensitivities previously observed in the AlaRS C666A mutant. Statistical significance was determined by one-way ANOVA with Tukey post-hoc comparison (* *p* < 0.0005).

**Figure 5 genes-11-01222-f005:**
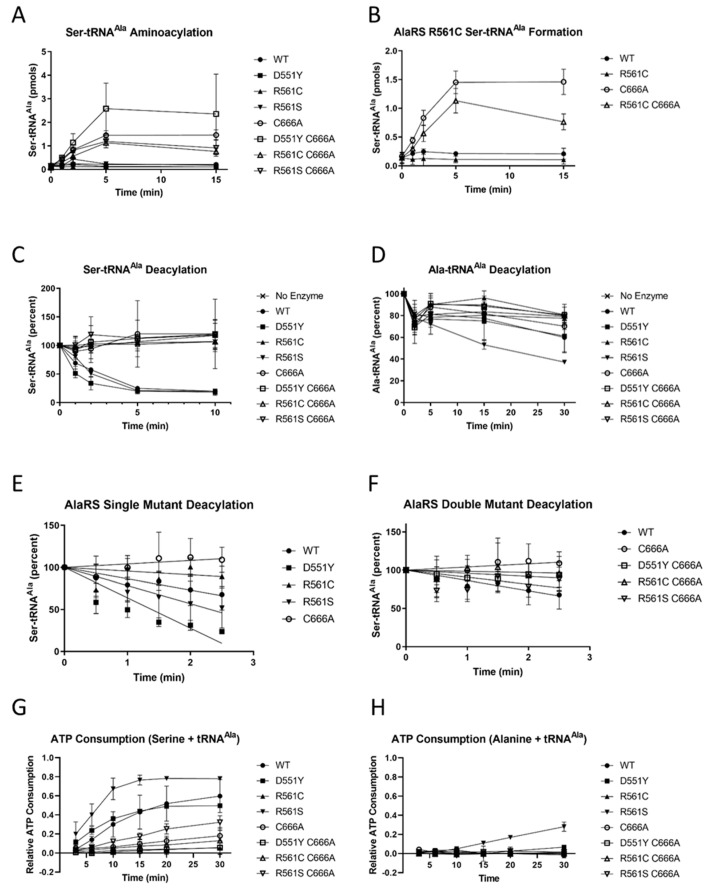
In vitro characterization of AlaRS C666A suppressors. The three identified AlaRS C666A suppressors were assayed for (**A**) mis-serylation activity and results from the (**B**) AlaRS R561C mutants were further analyzed. The proofreading activity of the enzymes were examined over a time course for (**C**) Ser-tRNA^Ala^ and (**D**) Ala-tRNA^Ala^ substrates. Furthermore the initial rates of Ser-tRNA^Ala^ deacylation were monitored for the (**E**) single mutants and (**F**) double mutants. Finally, ATP consumption was observed for reactions including tRNA^Ala^ and (**G**) serine or (**H**) alanine.

**Table 1 genes-11-01222-t001:** Steady state activation kinetics of AlaRS C666A suppressors. Steady state kinetic parameters of the AlaRS C666A suppressors for the activation of alanine and serine were determined using pyrophosphate exchange.

	Ala			Ser ^a^ (*k*_cat_/*K_m_*)	Specificity ^b^ (Ala/Ser)
	*K_m_*	*k* _cat_	*k*_cat_/*K_m_*		
	µM	min^−1^	min^−1^/µM	min^−1^/µM	
*E. coli* AlaRS	320 ± 150	1246 ± 105	3.91	4.7 × 10^−3^ ± 1.5 × 10^−4^	832
AlaRS C666A	263 ± 115	1010 ± 380	3.83	7.3 × 10^−3^ ± 5 × 10^−4^	525
AlaRS D551Y	260 ± 76	715 ± 220	2.76	6.9 × 10^−3^ ± 2.5 × 10^−4^	400
AlaRS D551Y C666A	217 ± 100	1096 ± 265	5.06	10.4 × 10^−3^ ± 14 × 10^−4^	490
AlaRS R561C	203 ± 68	1306 ± 345	6.45	6.7 × 10^−3^ ± 13 × 10^−4^	963
AlaRS R561C C666A	190 ± 30	819 ± 88	4.29	5.5 × 10^−3^ ± 19 × 10^−4^	780
AlaRS R561S	180 ± 15	1216 ± 125	6.70	8.2 × 10^−3^ ± 19 × 10^−4^	817
AlaRS R561S C666A	177 ± 37	791 ± 150	4.46	7.3 × 10^−3^ ± 5 × 10^−4^	611

a *K_cat_/K_m_* was estimated using subsaturating Ser concerntrations from the slope of the equation, V = *K_cat_*[E][S]/*K_m_*; b Measuted in *K_cat_/K_m_.*

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
