# Peer review of "Fine-Tuning of Alanyl-tRNA Synthetase Quality Control Alleviates Global Dysregulation of the Proteome"

_genes, 2020, doi:10.3390/genes11101222_

Round 1

Reviewer 1 Report

In this paper, the authors characterized residues within E. coli AlaRS proofreading domain that may function in quality control. They specified the second-site AlaRS suppressor mutationsbeside C666A, which is related to the mis-aminoacylated tRNA formation (Ser-tRNAAla), and confirmed that the defects in AlaRS proofreading lead to global dysregulation of the E. coli proteome, subsequently causing defects in growth, motility, and antibiotic sensitivity. The authors have been the specialists inthis research field and their results obtained by both in vivo and in vitro experimentsare extensive and clear. It should be highly noted that their strategy is quite logical and the process attaining to the conclusion is scientifically very sound.

(1)Although the authors compared the steady state kinetics of Ala C666A suppressors and discussed the natures of each variant, I am wondering the specificity value (Ala/Ser) of E. coli AlaRS in Table 1 (Ala/Ser = 832). The value of the N-terminal catalytic fragment of E. coli AlaRS (AlaRS441) is 310 (Guo et al. Nature 2009, 462, 808-812(the authors cited the paper as [27]).Is the difference of the valuesdue tothe Editing and C-Ala domains of AlaRS? The Ala/Ser value of AlaRS441(310) is close to that of AlaRS D551Y(400). As the authors wrote, if the D551Y substitution is likely productive for Ser-tRNAAla proofreading but not for preventing serine mis-activation, the results in Table 1 (PPi exchange using no tRNA)may be consistent with the low value of AlaRS D551Y (similar to that of AlaRS441). The authors should discuss the point.

(2)Although the authors provided the evidence of a “hyper-active”proofreading enzyme, probably based on elevated levels of diacylation andalsoATP consumption for cognate Ala-tRNAAla substrates, the definition of “hyper-active” should be explicitly described.

(3)Although serine and cysteine are isosteric, the effects of R561S and R561C variants are different. Could the effect of R561C be due to disulfidebond formation? I would be interested in the point. At least, the authors should comment on it, though in vitroexperiment in the presence of reducing agent, e.g., 2-mercaptoethanol or dithiothreitol for R561C variantis not necessarily mandatory.

(4)Line 22:proofreading → proofreading of Ser-tRNAAla

(5)Line 250:variants → variants except C666A

(6)Line 275: [13] → [13] (Figures 2A and 4A)

(7)Line 310: second-site → double mutant

(8)Line 369: R561C → R561S

(9)Line 412: AaRS → aaRS

(10)Line 494, 584, 624: Soll → Söll

(11)Line 646: Figure 1 → Figure3

Author Response

Although the authors compared the steady state kinetics of Ala C666A suppressors and discussed the natures of each variant, I am wondering the specificity value (Ala/Ser) of E. coli AlaRS in Table 1 (Ala/Ser = 832). The value of the N-terminal catalytic fragment of E. coli AlaRS (AlaRS441) is 310 (Guo et al. Nature 2009, 462, 808-812(the authors cited the paper as [27]). Is the difference of the values due to the Editing and C-Ala domains of AlaRS? The Ala/Ser value of AlaRS441(310) is close to that of AlaRS D551Y(400). As the authors wrote, if the D551Y substitution is likely productive for Ser-tRNAAla proofreading but not for preventing serine mis-activation, the results in Table 1 (PPi exchange using no tRNA) may be consistent with the low value of AlaRS D551Y (similar to that of AlaRS441). The authors should discuss the point.

Response: While there has yet to be any direct evidence that the editing domain or the C-Ala domain of AlaRS directly contributes to the increased specificity at the active site, there have been previous studies which show the role of the C-Ala domain in enzyme oligomerization [Banerjee B, Ganguli S and Banerjee B. (2020) Biophysical Characterization of Interaction between E. coli Alanyl-tRNA Synthetase with its Promoter DNA. Protein & Peptide Letters. 27, 1-14]. Based on these reports, we believe that the specificity data collected in the Guo et al. paper may have also been influenced by potential differences in enzyme oligomerization. While we cannot directly comment on the role of enzyme oligomerization and feel it is outside of the scope of this work, it is an interesting topic of discovery for future investigation.
It is curious that the specificity values of AlaRS D551Y and AlaRS441 are similar, we are cautious to over interpret the effects of the two enzyme constructs as the C-terminal truncation may have other effects on enzyme structure and activity that may not be representative of the effects of the D551Y mutant. We appreciate the reviewer’s comments and agree that the causes of similarities and/or differences between these reports are things to consider about the structure and function of this enzyme moving forward.

Although the authors provided the evidence of a “hyper-active” proofreading enzyme, probably based on elevated levels of diacylation and also ATP consumption for cognate Ala-tRNAAla substrates, the definition of “hyper-active” should be explicitly described.

Response: We appreciate the reviewer’s comment and modified the text. Lines 372-374 have been changed from “the AlaRS R561S variant appears to be hyper-active in AlaRS proofreading.” to “the AlaRS R561S variant appears to be hyper-active in AlaRS proofreading, as indicated by the increased hydrolytic activity.”

Although serine and cysteine are isosteric, the effects of R561S and R561C variants are different. Could the effect of R561C be due to disulfide bond formation? I would be interested in the point. At least, the authors should comment on it, though in vitro experiment in the presence of reducing agent, e.g., 2-mercaptoethanol or dithiothreitol for R561C variant is not necessarily mandatory.

Response:  We agree with the reviewers comment that the effect of the R561C mutation on the activity of the enzyme may different from the effect of the R561S mutation due to the formation of new or impairment to native disulfide bonds. We have modified the text to include this point. On line 368, the text now reads, “While it is yet to explored, the R561C substitution may cause disruption to wild-type AlaRS disulfide bond formation, leading to changes in enzyme activation and proofreading.”

Line 22:proofreading → proofreading of Ser-tRNAAla

Response: Line 22 has been corrected from “defects in AlaRS proofreading” to “defects in AlaRS proofreading of Ser-tRNAAla”.

Line 250:variants → variants except C666A

Response: While we appreciate the reviewer’s suggestion, we chose to not include “except C666A” because we wanted to emphasize that the second-site mutations arose in the context of C666A background. The single C666A mutation results in an editing-deficient enzyme with observable phenotypic defects and all of the identified suppressors alleviate the C666A-associated defects.

Line 275: [13] → [13] (Figures 2A and 4A)

Response: Line 275 has been corrected as per the reviewer’s suggestion.

Line 310: second-site → double mutant

Response: To maintain consistency throughout the manuscript, we have chosen to retain “second-site suppressors” instead of “double mutant”. As mentioned above, the term “second-site suppressors” emphasize that the suppressor mutations only arise in a C666A background.

Line 369: R561C → R561S

Response: The reviewer’s response is regarding the line “In contrast to observations studying the effect of the AlaRS R561C on aaRS discrimination, the AlaRS R561S variant appears to be hyper-active in AlaRS proofreading”. The use of R561C in this line is correct and is used to compare the effects of the R561S mutation in Section 3.7 (lines 371-388) to the effects of the R561C mutation in Section 3.6 (lines 361-370).

Line 412: AaRS → aaRS

Response: The sentence has been changed from “perturbation to AaRS fidelity” to “perturbation to aaRS fidelity”.

Line 494, 584, 624: Soll → Söll

Response: The appropriate changes have been made, as per the reviewer’s suggestion.

Line 646: Figure 1 → Figure3

Response: The figure legend has been amended.

Reviewer 2 Report

The proofreading activity of aaRSs to remove erroneously activated amino acids or charged tRNA is important for accurate translation and maintaining the integrity of the proteome. An alanyl-tRNA synthetase (AlaRS) mutant defective in this activity and having a C661A substitution in the proofreading domain can be responsible for impaired cell viability. In the present study, the authors identified three second mutations in the same domain that suppressed the growth defect due to the C661A mutation. They went so far as to conduct genomic engineering, to exclude the possible involvement of other genomic mutations in the isolated bacteria. Any of the suppressor mutations did not suppress the mistranslation of Ala to Ser in vivo, and why they alleviated the adverse effects of C661A remains unclear. Then, the authors thoroughly characterized the suppressor mutations in the in vitro activity of AlaRS, and showed that these mutations have distinct effects on the amino acid selectivity, proofreading activity, and trans-deacylation by AlaRS.

This study is complete with the detailed analyses of the suppressor mutations, and I believe is worth of publishing. It is still mysterious that the suppressor mutations revived cell viability, while allowing mistranslation in vivo. Perhaps, the underlying mechanism requires future studies to understand it.

One recommendation:

Some panels of Fig 5 are too crowded to see which mutants are represented by which lines. The lines might be in different colors for understanding.

Author Response

Some panels of Fig 5 are too crowded to see which mutants are represented by which lines. The lines might be in different colors for understanding.

Response: We appreciate the reviewer’s suggestion and recognize the visual strain caused by comparing so many data sets at once. While constructing these graphs we tried several approaches to showcase the data. We felt that the addition of color caused similar visual constraints when trying to differentiate between data sets. As all options led to similar challenges, we opted to keep the data black and white and modify the symbols. As most of the interpretation is comparing the data to either wild type AlaRS or AlaRS C666A, we felt that the data was still interpretable. In cases when greater resolution was required for examination of individual data, we included additional graphs (e.g. the data in Figure 5B is identical to the data in Figure 5A, but R561C is isolated to better visualize the changes). 

Reviewer 3 Report

The manuscript entitled "Fine-tuning of Alanyl-tRNA synthetase quality control alleviates global dysregulation of the proteome" identifies second-site AlaRS suppressor mutations that alleviate defects in growth, motility and antibiotic sensitivity that are found in E.coli strains presenting defects in AlaRS proofreading. This is a follow-up study of previous relevant findings by the same authors regarding the global dysregulation on the E. coli proteome due to AlaRS proofreading defects. The manuscript is well written and the methods are thoroughly described. The authors identify potential inconsistencies in some results and properly discuss them and adequately address them by performing additional experiments that strongly support the main conclusions of the manuscript.

It would be interesting to further explore the differences observed in the AlaRS R561C variant by trying to quantify errors in the entire E.coli proteome by mass spectrometry, performing protein synthesis rate measurements or even ribosome profiling, but these set of experiments would definitely make sense in a future study and are nor required for the publication of the present manuscript.

Minor revisions:

Figure 3 is wrongly identified as Figure 1.  

Author Response

Figure 3 is wrongly identified as Figure 1.

Response: The figure legend has been amended, as per the reviewer’s suggestions.